



# Intercomparison study and optical asphericity measurements of small ice particles in the CERN CLOUD experiment

Leonid Nichman[1], Emma Järvinen[2], James Dorsey[1, 3], Paul Connolly[1], Jonathan Duplissy[4], Claudia Fuchs[5], Karoliina Ignatius[6], Kamalika Sengupta[7], Frank Stratmann[6], Ottmar Möhler[2], Martin Schnaiter[2] and Martin Gallagher[1]

[1]{School of Earth, Atmospheric and Environmental Sciences, University of Manchester, Manchester M13 9PL, UK}

[2]{Institute of Meteorology and Climate Research, Karlsruhe Institute of Technology, Postfach 3640, 76021, Germany}

[3]{National Centre for Atmospheric Science, Manchester, UK}

[4]{Department of Physics, P.O. Box 64, 00014 University of Helsinki, Helsinki, Finland}

[5]{Laboratory of Atmospheric Chemistry, Paul Scherrer Institute, Villigen, Switzerland}

[6]{Leibniz Institute for Troposheric Research (TROPOS), 04318 Leipzig, Germany}

[7]{University of Leeds, School of Earth and Environment, LS2-9JT Leeds, UK}

*Correspondence to:* Leonid Nichman (Leonid.Nichman@manchester.ac.uk)

**Abstract**. Optical probes are frequently used for the detection of microphysical cloud particle properties such as liquid and ice phase, size and morphology. These properties can eventually influence the angular light scattering properties of cirrus clouds as well as the growth and accretion mechanisms of single cloud particles. In this study we compare four commonly used optical probes to examine their response to small cloud particles of different phase and asphericity. Cloud simulation experiments were conducted at the Cosmics-Leaving-OUtdoor-Droplets (CLOUD) chamber at European Organisation for Nuclear Research (CERN). The chamber was operated in a series of multi-step adiabatic expansions to produce growth and sublimation of ice particles at super- and sub-saturated ice conditions and for initial temperatures of –30, –40 and −50 °C. The experiments were performed for ice cloud formation via homogeneous ice nucleation. We report the microphysical properties of small quasi-spherical ice particles in deep convection simulations and small hexagonal ice particles typical for in situ cirrus. Ice crystal asphericity and a degree of submicron complexity deduced from measurements of spatially resolved single particle light scattering patterns by the Particle Phase Discriminator mark 2 (PPD-2K, Karlsruhe edition) were compared with Cloud and Aerosol Spectrometer with Polarisation (CASPOL) measurements and images captured by the 3View Cloud Particle Imager (3V-CPI). Averaged path light scattering properties of the simulated ice clouds were measured using the Scattering-Intensity-Measurements-for-the-Optical-detectioN-of-icE (SIMONE) and single particle scattering properties were measured by the CASPOL.

We show the ambiguity of several optical measurements in ice fraction determination of homogeneously frozen ice, in the case where sublimating quasi-spherical ice particles are present. Moreover, most of the instruments have shown a rather low sensitivity to the crystal complexity for small ice cloud particles that were grown under typical atmospheric conditions. Bulk averaged path depolarisation measurements of these clouds showed higher correlation to single particle measurements at high concentration and small diameters of cloud particles. These results have implications for the interpretation of atmospheric measurements and parametrisations for



modelling, particularly for low particle number concentration clouds. This ensemble of optical instruments,
using both averaged path and single particle detection presented here, in conjugation with the CLOUD chamber,
reveals the possible discrepancies in comparisons of airborne and remote sensing measurements.

## 1    Introduction

One of the first attempts to distinguish ice particles from water drops in the atmosphere was made almost 70
years ago in the Thunderstorm project (Byers and Braham,1948) during which it was noted that ice particles
produce different sound than water drops when they impact the canopy of the aircraft. Since then, there have
been many developments of airborne instruments for the measurement of cloud microphysical properties.
Wendisch and Brenguier (2013) compiled a comprehensive list covering 48 different instruments, many of
which are historical, but recently there have been several new developments, e.g. Abdelmonem et al. (2016) and
Baumgardner et al. (2014). Many of the current techniques however are technological improvements on
previous instruments originally developed and flown in the 1970's. An ongoing problem is the in situ
measurement of concentrations of small ice crystals < 100 μm in size. Accurate measurements of ice crystal size
distributions are necessary for evaluation of ice cloud radiative effects, development and evaluation of remote-
sensing algorithms, evaluation of aerosol impacts, and ultimately correct representation of ice clouds in climate
models (Jensen et al., 2009).
This microphysical information is also important in research of early ice formation when initial ice particles in
low concentrations push the sampling volume limits of many instruments (Johnson et al. 2014). Optical methods
are preferably employed both for remote sensing of clouds and for in situ single particle measurements. For the
detection of particle shape and structure, the scattering intensity of single particles is most commonly used. This
technique is compared here with the mean scattering intensity from ensemble measurements, where shape
information is averaged due to different orientations of the particles in the measuring volume (Sachweh et al.
23 1999).

The initial shape of ice particles may be indistinguishable from water droplets. Optically ambiguous shapes of
liquid and solid cloud particles such as water, frozen droplets and quasi-spherical ice (Gayet et al., 2012;
Järvinen et al., 2016c) may be detected simultaneously in the troposphere. Some of these particle phases coexist
for long periods of time e.g. in long-lived mixed-phase stratiform layers (Korolev and Isaac, 2003a). The
resolution of most optical probes, coupled with coincidence problems, prevents a clear determination of particle
shape for particle sizes smaller than 100 μm in such clouds. In glaciated clouds, Cober et al., (2001) applied
geometric formulas to 2D images, identifying between 5 and 40 % of them as circular. Spherical particles were
observed in large numbers by Korolev and Isaac (2003b) even in clouds sub-saturated with respect to water.
Moreover, Lawson et al., (2006) reported that particles < 50 μm account for 99 % of the total number
concentration, 69 % of the shortwave extinction, and 40 % of the mass in mid-latitude cirrus. As a result of this
shape ambiguity and low resolution of small sizes, our fundamental knowledge of small cloud particle
microphysics is far from being complete.
Large super-cooled water droplets up to 5 mm in diameter exist only at warmer ambient temperatures, but
smaller cloud droplets may frequently exist in a super-cooled state down to –20 °C, and less frequently as low as
the homogeneous freezing level (Elliott and Smith, 2015; Rosenfeld and Woodley, 2000). Furthermore, very



small super-cooled water droplets may stay in a metastable liquid condition down to –40 °C (Korolev et al.,
2003a). Pilots often reported deviating around convective clouds due to the danger of ice accretion of super-
cooled droplets (Jeanne et al., 2006). Therefore, an inaccurate classification of spherical shapes may directly
affect the routes and costs of commercial flights (Gallagher et al., 2016).
Frozen droplets are an important feature of mid latitude anvil cirrus. In fact, frozen droplets and frozen droplet
agglomerates are a dominant particle type also in higher anvil outflow clouds (Stith et al., 2014; Järvinen et al.,
2016c). Frozen droplets could also be responsible for first ice initiation in deep convective clouds (Taylor et al.,
2016). Although frozen droplets are frequently measured, our understanding of the microphysical and optical
properties of these quasi-spherical ice particles is somewhat vague. The process by which frozen droplets are
formed can play an important role in their morphology. Microscopic structures, like surface roughness, as well
as detailed information on the aspect ratios of the frozen droplets found in clouds are key variables required to
determine the optical parameters that are included in the modelling and prediction of the climate effect of these
cloud systems.
In addition, quasi-spherical ice shapes are common in cirrus. Luebke et al. (2015), Garrett et al., (2005) report
the presence of many quasi-spherical ice particles in cirrus, especially at the smaller sizes. Quasi-spherical ice
prevails also in contrails at low temperatures below about –55 °C. In the core of the contrail, high crystal
concentrations reduce the vapour density to saturation causing the ice particle to retain a nearly spherical shape
(Lawson et al., 1998; Lynch, 2001). Contrail cirrus cover is small compared to natural cirrus; nonetheless, they
still have a climatic importance with the constant increase in jet aircraft traffic (Stordal et al., 2005; Irvine and
Shine, 2015).
Cloud particles measurements on aircraft campaigns inherently suffer from limited spatial coverage and limited
instrument sampling volumes. Cirrus clouds do not have an obvious formation stage, and therefore it is not
possible to reliably position a research aircraft in their development stage (Lawson et al., 2006). Remote sensing
provides averaged features but is insensitive in case of subvisual or contrail cirrus. Conversely, chamber
experiments provide a well-controlled and pristine environment for simulations and instrumental comparison,
although the role of the ice nucleation process in the atmosphere may change with time through the life cycle of
a convective cloud for example, and is strongly influenced by the environmental airflow (Heymsfield et al.,

28 2005).

Our chamber campaign investigating the homogeneous freezing process relevant to the upper region of deep
convective clouds and in situ formed cirrus in pristine environments was conducted at the European
Organisation for Nuclear Research (CERN) in 2013, hereafter referred to as CLOUD 8. The goal of the
experiments presented here was to complement and extend the results previously obtained in the Aerosol
Interaction and Dynamics in the Atmosphere (AIDA) chamber with similar instruments (Järvinen et al., 2016c;
Schnaiter et al., 2016) such as observation of morphological features and confirmation of a possible pathway for
quasi-spherical ice formation which affects growth and sedimentation mechanisms of ice in clouds.
Additionally, a comparative analysis of four optical probes is reported in this paper to provide clarification of
optical measurements in several respects: single particle versus averaged path optical measurements,
polarisation measurements versus depolarisation and asphericity derivation i.e. using scattering patterns of the
near-forward scattered light in the Particle Phase Discriminator mark 2 (PPD-2K, Karlsruhe edition), single
particle polarisation properties in the Cloud and Aerosol Spectrometer with Polarisation (CASPOL), and image



analysis in the 3View Cloud Particle Imager (3V-CPI). We then use the asphericity to determine the ice fraction
in a cloud by prescribing an aspherical shape for all the ice particles, and hence assume that ice fraction is
equivalent to an aspherical fraction.
**2      Methodology**
**2.1      The CLOUD chamber**
The chamber facility at CERN is described in detail by Duplissy et al. (2016), Kirkby et al., (2011) and Guida et
al., (2013). The expansion system installed at the Cosmics Leaving OUtdoor Droplets (CLOUD) chamber
allows production of relatively high cooling rates, above 5 °C min[-1], compared to the AIDA chamber, where
maximum cooling rate of 4 °C min[-1] is typically achieved (Möhler et al. 2006; Järvinen et al., 2016c). Stronger
cooling rates will activate a higher fraction of the aerosol by driving higher peak super-saturation. Since the
liquid water content that freezes does not vary with updraft strength, the freezing of more numerous droplets in
the faster updrafts simply produces smaller ice particles. This is clearly shown by Ackerman et al. (2015) where
ice particle mass distributions in homogeneous freezing for stronger updrafts produce substantially smaller ice
particles and greater ice water content. Schnaiter et al. (2016) further showed that high ice particle growth rates
also enhance the formation of small-scale complexity, such as ice particle surface roughness. Following the
procedure suggested by Schnaiter et al. (2016), we have simulated similar conditions for the derivation of
aspherical fractions and instrumental inter-comparison in the CLOUD chamber, where ice particles are
sequentially sublimated and then grown under different supersaturated conditions. The multistep adiabatic
expansion mechanism has allowed the regrowth of ice after sublimation as will be explained in the next section.
**2.2      Overview of the homogeneous freezing experiments**
In the cloud chamber experiments, we have simulated some of the homogeneous freezing processes taking place
in the deep convective cloud systems i.e. with updraft velocities up to 5 m s[-1] with corresponding cooling rates
up to 5.8 °C min[-1]. In the following sections we present the evolution of the ice particle shape and small-scale
complexity upon freezing, sublimation and re-growth periods from selected representative individual
experimental runs. Overall, all the results from repeated individual experiments agreed well with each other. A
representative list of the conducted experiments can be found in Table 1. The technical description of the
CLOUD chamber pressurisation, CCN injection and expansion of the air volume in a multi-step regime is given
in detail elsewhere (Nichman et al., 2016; Duplissy et. al., 2016; Guida et al., 2012, 2013) and will be briefly
described here.
The homogeneous freezing experiments were started in a pressurized chamber volume, at 123.3 kPa, with a
CCN injection. The sulphuric acid solution droplets, used as CCN, were generated in a sulphuric acid generator
consisting of heated sulphuric acid reservoir and airflow past the reservoir. A more detailed explanation of the
generation method can be found in Wagner et al. (2008). By varying the duration of the sulphuric acid injection,
we controlled the number concentration of the sulphuric acid droplets, and by adjusting the temperature of the
sulphuric acid reservoir and the airflow rate through the heated reservoir we controlled the mean size of the
aerosol particles.





The chamber is surrounded by an insulated thermal housing which allows a precise regulation of the
temperature with stability within 0.1 °C. The in situ temperature values are measured close to the centre of the
chamber, at 1.2 m distance from the walls using a PT100 temperature sensor (Duplissy et al., 2016; Dias et al.,
2016). For pressure monitoring in the chamber, we used the VEGABAR 53 process pressure transmitter,
VEGA.
At the beginning of most experiments, we generated low concentrations (~ 100 cm$^{-3}$) of sulphuric acid aerosol.
At these concentrations, all seed aerosols will act as CCN at almost the same time and further homogeneous
nucleation and growth of the ice crystals would occur upon further expansion cooling. All the experiments were
initiated slightly below ice saturated conditions at temperatures near –30, –40 and –50 °C. In the experiments
starting at –30 °C we cooled the chamber air by expanding the volume until super cooled liquid droplets were
formed. The droplets were grown by further cooling until ice started to form by homogeneous freezing of the
super-cooled water droplets. The ice particles then grew until the expansion was stopped and the formed ice
crystals started to sublimate under ice sub-saturated conditions, induced by an increase of temperature due to the
heat flow from the warm chamber walls. A second step in the expansion profile allowed the regrowth of the
sublimating particles. The experiments at –40 and –50 °C were started similarly by cooling the chamber volume
until the first ice particles were formed by deposition nucleation. After the first ice particles were detected and
had grown in diameter (Table 1), we proceeded to the next step of the multistep expansion profile as discussed
above.
**2.3    Cloud probes**
**2.3.1    PPD-2K**
The classification of cloud particles by the PPD-2K is based on a spatial analysis of high resolution intensity
patterns of single particles in the 5 to 26° forward angular range. In the scattering patterns of (spherical) droplets
we normally observe concentric rings at angular positions corresponding to the maxima intensities given by
Mie-theory. We used computerised discrimination of images with concentric rings from images without the
rings based on variance calculation of the image pixels along the polar integrated azimuthal intensity profiles
(Vochezer et al., 2016). Aspherical fractions were determined by applying a threshold variance value of $10^{-5}$.
Images with low variance corresponded to concentric rings and were classified as spherical (e.g. droplets).
Similarly, in the case of ice particles with mean variance below this value the particles were classified as
spherical.  Ice habits e.g. columns and plates have characteristic scattering patterns which allow classification of
the detected particles. More technical details are described in Vochezer et al. (2016).
**2.3.2    SIMONE**
The averaged path SIMONE-Junior (Järvinen et al., 2016b), was installed in the chamber for bulk depolarisation
measurements. This instrument is comparable to a lidar and is used to detect phase-transitions in aerosol, cloud
particle ensembles, and to investigate the microphysical properties of clouds.  The instrument projects a 552 nm
polarised (e.g. perpendicular, parallel and circular) light beam and detects from a volume of a few cubic
centimetres. Unlike a lidar measurement, parallel and perpendicular components of the backscattered light are
measured around the detection angle of 176°, at a very confined angular range with an acceptance angle less
than 0.8 mrad. The linear depolarisation ratio is zero for spherical particles and non-zero if particles' shape



deviates from a sphere, thus a detection of the bulk cloud phase is possible. Forward scattering intensity is
measured at 4°. The operation of the SIMONE in the CLOUD chamber is described in detail by Järvinen et al.
(2016a). The basic instrument concept and data interpretation in case of chamber ice clouds is detailed in
Schnaiter et al. (2012).
### 2.3.3  Airborne probes
**CASPOL**
The CASPOL installed in the chamber was part of the Cloud, Aerosol, and Precipitation Spectrometer (CAPS,
Droplet Measurement Technologies), an instrument commonly used on aircraft for cloud microphysical
measurements (e.g. Baumgardner et al., 2001; Johnson et al., 2012; Luebke et al., 2015). The CASPOL relies on
incident laser-scattering by single particles. The collecting optics guide the light scattered in the 4 to 12°
subtended cone into a forward-sizing photodetector. This light is measured and used to infer particle size.
However, the CASPOL is calibrated with spheres and aspherical particles are mis-sized (Borrmann et al., 2000).
We estimate the sizing error would normally be of the order of the size bin width.
The backscatter detector measures the scattered light cone subtended by the angles 168 to 176°. Additionally,
this version of CASPOL measures the polarised fraction of the backscattered light in the orthogonal plane for
the first 292 particles s$^{-1}$ (Droplet Measurement Technologies Manual, 2011). This functionality allows
discrimination of aspherical particles in the 0.51 - 50 μm range. For spherical particles, typically droplets, the
polarisation of the incident light will be preserved and the orthogonal polarisation in the back-scatter will
generate nearly zero signal. Depending on the asphericity of the particles, there will be increased signal in the
backscatter polarised detector. An increase in size with decrease in polarisation in CASPOL at temperatures
below the frost point will mean that ice is sublimating and becoming more spherical (Jensen et al., 2010). The
classification of droplets and ice in CASPOL data analysis is primarily based on polarisation threshold which
needs to be determined from laboratory experiments (Nichman et al., 2016).
**3VCPI**
Another aircraft mounted instrument is the Cloud Particle Imager which can image and count particles in the
size range of 15–2500 μm, with the images having a nominal 2.3 μm resolution. The newer 3V-CPI (SPEC Inc.)
is essentially a 400 frame per second CPI probe integrated with a 2D-S probe. CPI obtained information covers
particle size (including area and volume) and ice habit classification (Heymsfield et al., 2010). Complementary
size distributions and concentrations data are obtained by the 2D-S. The 3V-CPI is especially suitable for use in
ice and mixed phase clouds (Lawson et al., 2003; Gayet et al., 2012; Stith et al., 2014). Each of the 2.3 μm
resolved surface images captured by the CPI can be fitted to a circle function to determine the roundness of the
particle (Korolev and Isaac, 2003b). Temporal changes of roundness can be used to calculate the mean non-
round (aspherical) concentration fraction. However, the roundness parameter for smallest detected particles of
10 μm optical diameter, have the largest uncertainty as will be discussed in Sect. 3.2.



# 3    Results and discussion
## 3.1    Experimental description
### 3.1.1    Ice nucleation and regrowth
The air pressure and mean temperature profiles for a typical expansion procedure in accordance with Schnaiter
et al. (2016) are presented in Fig. 1a. The expansion starts first with a slow pressure decrease to create water-
supersaturated conditions inside the chamber and to form a cloud of super-cooled droplets. The expansion rate is
increased towards the end in order to achieve ice-supersaturated conditions in a short time period and to nucleate
the ice almost simultaneously. The overall cooling rate during this expansion was –4.9 °C min$^{-1}$. The PPD-2K
measured the size distribution during the expansion (Fig. 1b). The cloud period with super-cooled droplets lasts
only a few seconds and followed by almost immediate and fast ice formation. The depolarisation signal
measured by the SIMONE increases promptly after the increase in the forward scattering signal indicating the
short droplet period followed by the fast formation of ice (Fig 1c). The expansion is then stopped at ~ 1 min
(Fig. 1a).
After the first step of the expansion, the initial temperature of the air volume is slowly restored by the heat flux
from the warmer chamber walls, thus creating sub-saturated conditions inside the chamber. This warming leads
to the sublimation of ice crystals and the observed changes in their microphysical properties. A re-growth of the
sublimating ice crystals is initiated at ~ 11 min, when the pressure decreases from 111.3 to 101.8 kPa. The
depolarisation signal increases once again during this step and reaches slightly higher levels (0.34) than in the
first step (0.26), together with an increasing noise level due to the low number concentration. There is also a
small increase in the forward scattering but this is much lower than in the first step of the expansion due to ~5
fold decrease in the concentration. We assume a complete glaciation in the first step without any significant
reactivation in the number concentration during the regrowth period (Fig 1c).
### 3.1.2    Size range overlap
In this comparison, the overlapping size range of PPD-2K and CASPOL for measurements of small ice particles
is 7 – 50 μm (Fig. 1b). However, only 41 % of the particle-by-particle (PBP) polarisation data in CASPOL at –
30 °C were from particles larger than 7 μm. At lower temperatures the particle size distribution (PSD) is shifted
towards the grey area in Fig. 2, below the PPD-2K size cut-off. The fraction of CASPOL PBP polarisation data
points from particles > 7 μm at lower temperatures was even lower: 24 % (–40 °C), 32 % (–50 °C), thus, most
of the particles in the cloud that produce the polarisation data in the CASPOL are small, < 7 μm, while in the
PPD-2K data 100% of the analysed particles are > 7 μm.
The size segregated aspherical fractions as measured by the PPD-2K in the overlapped size region are presented
in Fig. 3. At –30 °C, the large ice particles reach 20 μm in diameter during the first step of the expansion and
grow up to 35 μm during the second step, within the detection range of CASPOL. Ice particles formed under
different temperature regimes would have a different morphology. At cirrus temperatures below –40 °C, the ice
particles form directly from the vapour phase via deposition nucleation (Figs. 3b, 3c), a different formation
pathway compared to ice formation through the liquid phase at –30 °C (Fig. 3a). At the final step of the
expansion at –30 °C, during the sublimation period, the aspherical fraction is extremely low due to sphericity
ambiguity as will be discussed in Sect. 3.3. For better statistical characterisation we achieved longer cloud life



time at lower temperatures, up to 45 min. However the measurements of PPD-2K at these temperatures were
somewhat incomplete, missing the smaller sizes and hence the initial steps of cloud particle formation and
growth especially during the first step of the expansion (Figs. 3b, 3c). The temperatures profiles of runs:
1276.05 (–40 °C), and 1298.12 (–50 °C) are shown in Fig. S1.
**3.1.3    Column fraction**
The ice fraction contains ice habits such as plates and columns in all the regrowth experiments discussed here
(Table 1). The largest fraction detected by PPD-2K at different temperatures was composed of ice columns as
shown in Fig. 4. In the first part of experiment 1292.01 (–30 °C) the frozen droplets are grown at lower
temperature and higher super-saturation than in the second sublimation period of this experiment (Fig. 1a),
leading to a formation of complex particles (Sect. 3.3). In the regrowth period, the temperature drop and super-
saturation conditions are more moderate and we observe the formation of columnar ice particles. The columnar
shape is not preserved and the ice particles sublimate to their underlying spherical core as seen in Fig. 3a. The
largest column fractions were measured at the lowest temperature –50 °C (Fig. 4c). Although we know that
diffraction instruments (e.g. SID-1, SID-2, SID-3, Hirst et al., 2001; Cotton et al., 2010; Vochezer et al., 2016)
suffer from major coincidence errors at high concentration and mixed phase clouds, in these chamber
experiments we did not observe coincidence errors.
**3.2    3V-CPI image analysis of ice particles**
High concentrations of particles grew above 20 μm in diameter (Fig. 1b), thus allowing their detection with the
3V-CPI instrument. Quasi-spherical or quasi-spheroidal small particles were identified from the CPI images
(Fig. 5). The CPI imager is triggered by the 2D-S component for particles larger or equal to 10 μm as described
in Sect. 2.3.3. The image analysis can provide the roundness of the particles. Due to the larger error in small
sizes, Korolev and Isaac (2003b) have considered the roundness of the particles with diameter larger than 20 μm
is appropriate. Connolly et al. (2007) have included the roundness of smaller particles of 10 μm in diameter in
their analysis using size and shape corrections based on tests with ice analogues to the instrument's depth of
field. Emersic et al. (2015) chose a roundness threshold of 0.9 for phase discrimination of particles larger than
35 μm. The exact definition and calculations of roundness are described in detail in the papers above. In this
analysis we consider particles in the range of 20–50 μm for broader coverage of the CASPOL and PPD-2K size
ranges while constraining the uncertainty in the roundness parameter. Here, the threshold for phase
discrimination by roundness was set to 0.9.
Analysis of a large dataset of CPI images by Korolev et al., (2003c) showed that in glaciated clouds a large
fraction of particles with diameter < 60 μm do indeed have a quasi-spherical compact shape. Korolev and Isaac,
(2003b) noted that the question of spherical ambiguity remains due to optical limitations of the instruments.
Despite the limitations of size range and resolution, Fig. 6a shows an increase of the non-round image fraction
during the growth periods of the ice particles in our chamber experiments.
**3.3    Aspherical fractions measured by PPD-2K, CASPOL, 3V-CPI**
In airborne measurements, ice fractions are commonly derived from the optical asphericity of the particles (Sect.
1). Here we compare PPD-2K and CASPOL single particle measurements to examine their ability to distinguish



between droplets and ice particles in different ice nucleation modes. In experiment no. 1292.01 (–30 °C), all the
droplets freeze simultaneously, almost immediately after their formation, concluding the duration of the pure
liquid cloud in the order of seconds (Fig. 6a). Promptly after freezing we measure a high ice (aspherical)
fraction (~100 %), as expected. The freezing onset is detected by the increase in the depolarisation signal from
the SIMONE, and proceeds with an increase in 3V-CPI non-round fraction (Sect. 3.2), both indicating the
presence of non-spherical particles. In the sublimation period, at 4 min, we see the reversed transition in the
aspherical fraction; we start to detect more spherical particles that are classified as liquid droplets according to
the thresholds used to classify ice (Sect. 2.3; Nichman et al., (2016)). The depolarisation finally decreases (Fig.
6a), and particles are no longer detected by the 3V-CPI due to their decrease in size below the threshold. The
size segregated ice fraction for this experiment is shown in Fig. 3a.
During the sublimation periods the aspherical fraction decreases implying increasing sphericity of the particles
(Fig. 6a). However, once full glaciation was observed, the liquid phase cannot subsequently exist at the ambient
chamber temperature; below –30 °C. Therefore, the nearly spherical particles observed (4–10 min, 19 min) are
spherical ice and not liquid water droplets. In atmospheric measurements, such an aspherical fraction would
normally be converted into an ice fraction. In this cloud simulation, at the end of the sublimation period, both
instruments misinterpret the total ice fraction as spherical-liquid by 60%.
In experiments 1276.05 (–40 °C) and 1298.12 (–50 °C) (Figs. 5b, 5c), we observe a lower aspherical fraction
measured by the CASPOL at a lower temperature. Polarisation data analysis therefore suggests that ice particles
smaller than 7 µm are more spherical at lower temperatures (Fig. 6c red line) and they are more abundant (Sect.
3.1.2). Furthermore, the aspherical fraction for all particles detected in the PBP mode in CASPOL at –50 °C
follows the SIMONE depolarisation time series while the aspherical fraction of CASPOL >7 µm subgroup is
higher and increases towards the end of the expansion (Fig. 6c). The size dependence of these two polarisation
detection techniques is demonstrated in Sect. 3.4. However, scattering patterns detected by the PPD-2K showed
a 100% aspherical fraction in both experiments with vapour formed ice crystals.
Similar discrepancies in aspherical fraction measurements by PPD-2K and CASPOL were shown already by
Järvinen et al., (2016c) for ice particles formed via homogeneous nucleation and via deposition nucleation on
mineral dust at –30 °C. The asphericity of the particles significantly differs for ice formed through the liquid
phase and ice formed through the vapour phase. These discrepancies (Figs. 5b, 5c) at lower temperatures can be
partially explained by the decrease in the number of particles measured in every second and therefore large
standard deviation in aspherical fraction calculation i.e. 19 % for particles > 7 µm at –40 °C and 37 % for
particles > 7 µm at –50 °C. However, the smaller size of particles at –40, –50 °C is the main cause to reduced
sensitivity of the polarisation measurements in respect to aspherical features as will be explained in the next
subsection.
**The impact of small-scale surface complexity on phase discrimination**
The resolution of instrumentation employed in atmospheric measurements, commonly, is not sensitive enough
to image the surface microstructural features of the ice crystals. The resolution of the widely used CPI probe is
around 2 µm, i.e. in the same range as the smallest droplets that are frozen into ice crystals and significantly
larger than the size of the ice crystals' surface anomalies. Although these anomalies, like roughness and stepped





hollowness of the crystal, do not significantly contribute to the mass distribution, they can significantly alter the
light scattering properties of the ice crystals, as discussed in the introduction.
We have analysed the scattering patterns of the PPD-2K instrument to determine the surface features of
individual ice crystals as described in detail in Schnaiter et al. (2016). This instrument is sensitive to features
that are on the order of the wavelength used, 532 nm. The particle's surface complexity or non-uniformity
manifests itself as speckles in the diffraction patterns (Järvinen et al., 2016c), where the analysis of spatial
uniformity of the scattered light intensity with the Grey-level Co-Occurrence Matrix (Schnaiter et al., 2016)
indicates the small-scale complexity of an ice crystal. The so called k-value defined as a complexity parameter
by Schnaiter et al. (2016) can reflect the physical complexity in a higher k-value. However we should emphasize
that at present it is not possible to quantitatively relate this value to an actual degree of complexity or surface
uniformity of the particle. We also note here that although a k-value can be calculated from the PPD-2K
instrument, the k-value is not calibrated for this instrument. Therefore, in this study we can only conclude about
the relative variations in this complexity parameter.
Figure 7a shows the size segregated k-value in experiment no. 1292.01. The experiment starts with a period of
super cooled liquid droplets, where scattering patterns of concentric rings are observed (scattering pattern A in
Fig. 7a). The freezing of the droplets takes place almost simultaneously and all the freshly frozen droplets
exhibit surface features i.e. the PPD-2K scattering patterns are speckled without the normal concentric ring
features (scattering pattern B in Fig. 7a). Seemingly the frozen droplets develop a frost layer (Järvinen 2016c)
on their surfaces during the freezing and initial growth. However this layer was too thin to be detected by the
other instruments.
During the sublimation period, the complexity of particles noticeably decreased (Fig. 7b). The smooth frozen
droplets were found at smaller sizes compared to the frozen droplets with small-scale complexity. The
sublimation of the frost can be seen in scattering patterns C and D (Fig. 7), where the diffraction pattern D
cannot easily be distinguished anymore from that of a liquid droplet, i.e. the frozen droplet in question is almost
a perfect sphere.
The re-growth of the sublimating frozen droplets, achieved by again starting expansion leading to ice super
saturated conditions. The surface frost layer developed again immediately after super saturation is reached and
the median complexity parameter reaches its' highest value (Fig. 7b). Consequently, these particles are
immediately classified as aspherical. The observed complexity of pattern B reappears on pattern E i.e. on the
columns (Figs. 4a, 7). The re-grown ice crystals were again allowed to sublimate after the growth period.
Similar smooth frozen droplets were detected as in the first sublimation period, together with a decline of the
detected aspherical fraction.
The small-scale complexity significantly changes the scattering patters measured by the PPD-2K (Fig. 7).
Therefore, the observed speckled patterns can be easily classified as aspherical, which leads to a robust and high
ice fraction, if complex ice particles are present. Sublimating columnar particles that demonstrate lower
complexity (Fig. S2) are classified as aspherical in the PPD-2K. Only optically spherical sublimating frozen
droplets are misclassified as droplets by the automatic algorithm. However, with visual inspection it is still
possible to discriminate between a smooth ice sphere and a spherical liquid droplet (compare patterns A and D
in Fig. 7). However, other optical instruments (e.g. CASPOL) can misclassify complex ice particles if they are
too small, and underestimate the aspherical fraction if their occurrence is low.





### 3.4     Single particle polarisation and ensemble depolarisation ratios
This analysis aimed at improving our interpretation of the small ice particle polarizability and the comparison of
different instruments and their approaches to discriminate small liquid and ice phase cloud particles by
properties of the scattered light. One such property is the linear depolarisation ratio for parallel incident laser
polarisation $\delta_{||}$ which is defined as the perpendicularly polarised to parallel polarised ratio of the backscattered
light intensity $(I_\perp/I_{||})$. This ratio is frequently used in remote sensing (e.g. Burton et al., 2012; Petzold et al.,
2010). The linear polarisation ratio in CASPOL data analysis is defined as the fraction of the perpendicularly
polarised backscattered light from the total backscattered intensity (Dpol/Back) as previously reported (Glen
and Brooks, 2013) and is used for ice fraction derivation (Nichman et al., 2016). The particle detection method
and the measured polarisation components are not the only dissimilarities in these instruments, they also operate
at different wavelengths, have slightly different collection angles and two orders of magnitude difference in the
sample volume (see Sect. 2.3).
Depolarisation ratio measured by the SIMONE and the polarisation ratio measured by the CASPOL were
plotted against each other with 1 s temporal resolution for run no. 1292.01 (high concentration and small
diameter), no. 1291.12 (low concentrations and big diameter), no. 1291.07 (high concentration and small
diameter) and no. 1298.2 (low concentration and small diameter) (Table 1) are shown in Fig. 8. The dimensions
of the marker reflect the number concentration, with the highest at 56 cm$^{-3}$. The CASPOL polarisation ratios
presented here are based on 1 s averages for all particles larger than 3 μm. Small aerosol particles (< 3 μm) were
detected during background measurements; after CCN injection and before the expansion. These small particles
i.e. non-activated aerosols were excluded from further analysis.
The Lowest correlation was found at low concentration, where the detected polarisation is higher than the
depolarisation. This is partially due to the averaging deviation at low concentration within the large sample
volume of the SIMONE instrument. Nonetheless, in several cases a surprisingly reasonable correlation is
observed (Fig. 8a, 8b). Generally, we have found that higher correlation is observed between the different
instruments in cases with high concentration and small diameters of cloud particles ($R^2 \sim 0.35$). Low correlation
($R^2 \sim 0.01$) is observed in cases with low concentrations and larger sizes of cloud particles.
### 3.5     Implications to atmospheric measurements
The classification of quasi-spherical ice as liquid droplets is posing a problem in atmospheric measurement and
especially in mixed-phase clouds, where the ice fraction calculations can be affected by the misinterpretation
presented in this paper. In these chamber experiments, we know that all the particles in the sub-saturated
conditions are in the ice phase, which allows us to use the spherical classification method to distinguish
spherical ice particles in cold clouds. In the atmosphere, there are additional possible crystal rounding
mechanisms e.g. equilibrium thermal roughening near 0 °C, a surface coating of solution, kinetic roughening at
high super saturations, and latent heat-induced melting of the surface during growth at high temperatures and
super saturation. In addition, frozen droplets retain their rounded appearance until sufficient growth occurs.
Therefore, it is not possible to infer sub-saturated conditions in the atmosphere merely by sampling rounded
crystals (Nelson 1998) and the measured ice fraction is prone to significant underestimation. In any case of
small quasi-spherical particles detection at sub-zero temperatures in the atmosphere, we recommend to compare
the data of more than one instrument. Ice fraction derived from CASPOL data can be compared to other



1. instruments with higher confidence when the PSD is fully covered by the overlapped size and concentration
2. range of the instruments with sufficient number of particles for ice fraction derivation and low standard
3. deviation. However, high concentration may cause coincidence and misestimation of the ice fraction.
4. The comparison of remote sensing and PBP measurements is not a straightforward process (i.e. bulk vs. single
5. particle and single complexity vs mixed-complexity ensembles of particles). Many single particle laboratory
6. techniques in particular have proven difficult to adopt when translated to real atmospheric environments. These
7. techniques often provide complementary data rather than comparable data (Lynch, 2001) and research in this
8. area continues. Based on our analysis, ensemble depolarisation measurements of cloud particles at certain
9. concentrations, sizes and atmospheric conditions can be comparable to single particle airborne measurements.

10. **4    Conclusions**

11. We have presented an instrumental setup for combined single cloud particle and ensemble measurements for
12. assessment of the relative optical ice and liquid responses in each case. The results were used to determine the
13. ice shape and small-scale complexity evolution during adiabatic expansion, sublimation and regrowth as well as
14. for potential impact on phase discrimination. We report observations of super-cooled and frozen droplets, small
15. ice habits and spheroids in a series of CLOUD chamber experiments at –30, –40 and –50 °C.
16. We have shown that the small quasi-spherical ice particles produced in the sublimation process exhibit a similar
17. optical behaviour to that of water droplets in the PPD-2K variance analysis and in the CASPOL polarisation
18. analysis for high PSD overlap at –30 °C. The analysis of the scattering patterns shows the similarity of the
19. spherical states and the difficulty in applying automatic phase discrimination. Therefore, observations of small
20. spheroids (< 60 μm) in sub-saturated conditions might be highly ambiguous. These results indicate that small
21. quasi-spherical ice misclassification might similarly concern numerous optical instruments, impactors and other
22. probes that were not examined here. Nonetheless, the scattering patterns differ for quasi-spherical ice and water
23. due to small deviations from sphericity. An increase of resolution in future versions of the optical instruments
24. might amplify this discrimination and reveal additional subtle features.
25. We have shown a chamber simulation of small-scale complexity evolution on a frozen droplet during an updraft
26. and in sub-saturated conditions. In regions with high concentration of small cloud particles (< 60 μm), the
27. observed differences in morphology will affect the observed radiative properties, growth mechanisms,
28. aggregation and charging in clouds. The aspherical fraction detected by the PPD-2K could be described with a
29. high degree of small-scale complexity which was undetectable by the other instruments. This complexity
30. measurement can be potentially calibrated in future experiments to derive the complex fraction of ice particles.
31. However, the complexity of ice particles smaller than 7 μm remains unclear.
32. We have presented polarisation measurements of airborne and laboratory-instruments in an expansion chamber.
33. We conclude that in these simulated atmospheric conditions the polarisation and depolarisation signal from
34. frozen droplets have higher correlation at higher concentrations of small particles and can be comparable above
35. certain concentration and size thresholds. These findings and the derived instrumental differences can be used in
36. the interpretation of atmospheric measurements of frozen droplets from remote and in situ, combined campaigns
37. as    well    as    a    pathway    for    further    research    and    development    of    these    instruments.
38.



*Acknowledgments*. We would like to thank CERN for supporting CLOUD with important technical and financial
resources. We express great appreciation for the CLOUD collaboration. This research has received funding from
the EC Seventh Framework Programme (Marie Curie Initial Training Network "CLOUD-TRAIN" no. 316662).
The SIMONE measurements were funded by the CERN CLOUD project. The PPD-2K instrument was fully
funded by the Deutsche Forschungsgemeinschaft within project SCHN 1140/2-1. The CAPS and 3V-CPI
instruments used in this work were supplied by the UK National Centre for Atmospheric Science.



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





1    **Table 1: List of experiments**

| Exp. number | CCN conc. [cm$^{-3}$] | Cooling rate [°C min$^{-1}$] | T start [°C] | Mean diameter [μm] | Mean dN/dlogD$_p$ [cm$^{-3}$] |
|---|---|---|---|---|---|
| 1276.05 | 220 | –5 | –40 | 7.9 | 48 |
| 1291.07 | 160 | –5.8 | –30 | 10 | 41.7 |
| 1291.12 | 110 | –4.8 | –30 | 15 | 30 |
| 1292.01 | 150 | –4.9 | –30 | 12 | 41.7 |
| 1298.12 | 110 | –2.1 | –50 | 9 | 6.5 |
| 1298.20 | 750 | –3.1 | –50 | 8 | 9.9 |



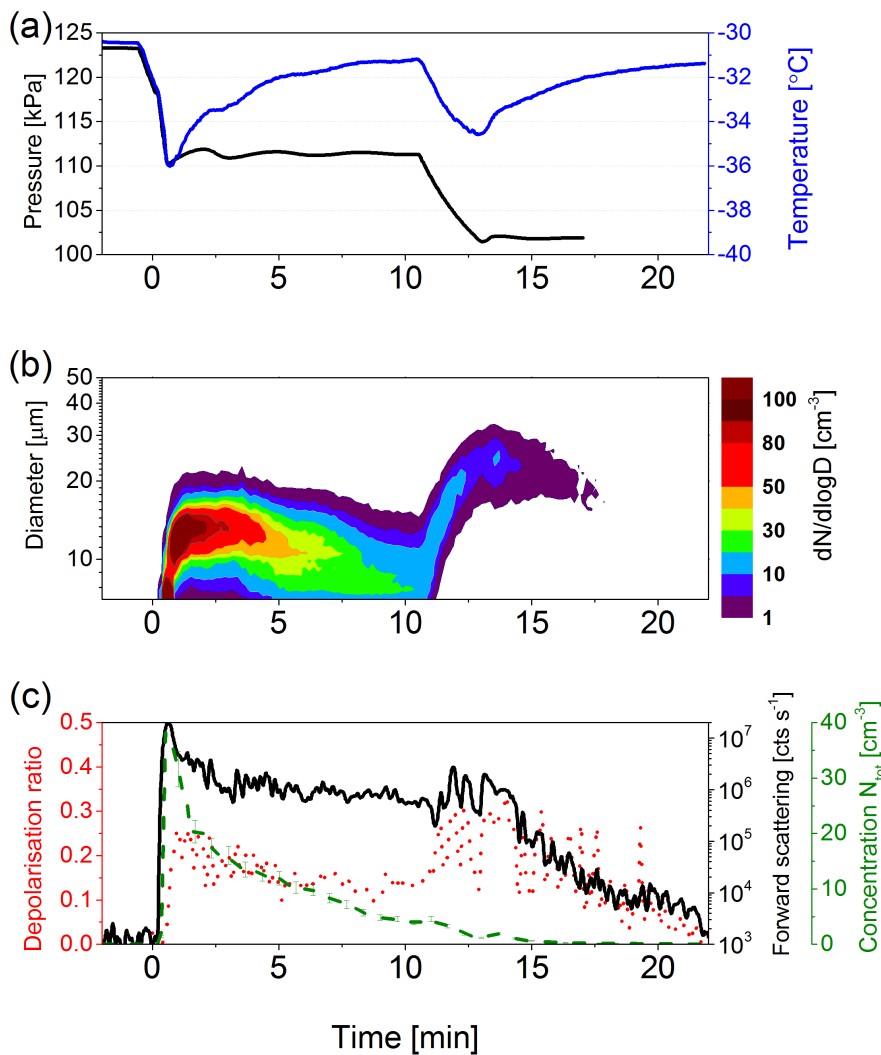

**Figure 1. Homogeneous ice nucleation and regrowth experiment no. 1292.01 (–30 °C). (a) The development of**

**pressure and temperature. Cloud forms at t=0 min, (b) The size distribution measured with PPD-2K, (c) SIMONE**

**measurements of the forward scattering intensity (black solid line) and depolarisation ratio (red dotted line) together**

**with the total number concentration measured by the PPD-2K (green dashed line).**



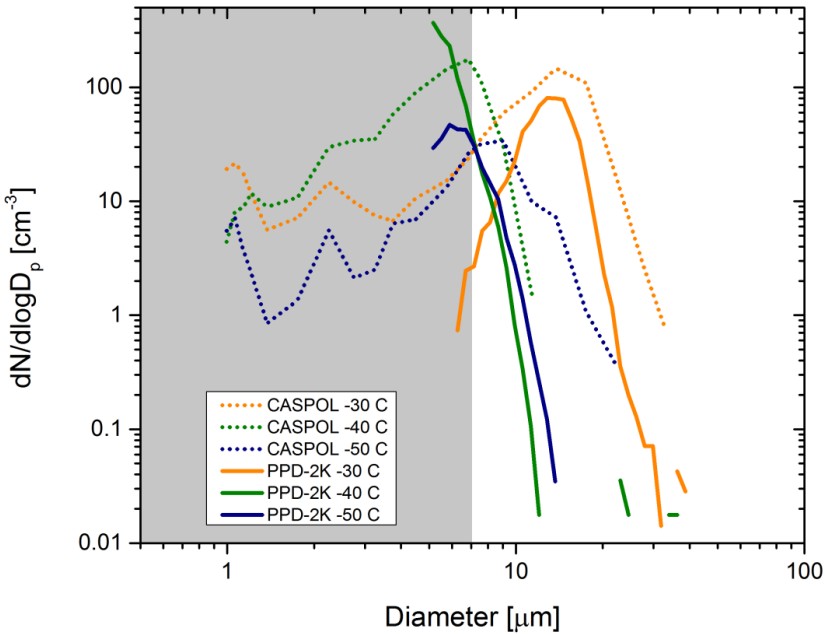

**Figure 2. Selected 1 min averaged particle size distributions for runs: 1292.01 (–30 °C), 1276.05 (–40 °C), 1298.12 (–50
°C). White area represents the overlap in the size range of PPD-2K and CASPOL. Grey area represents the particles
that are mostly present in the 292 PBP polarisation data points in a second. The aspherical fraction for comparison in
Fig. 6 is derived from the white area.**





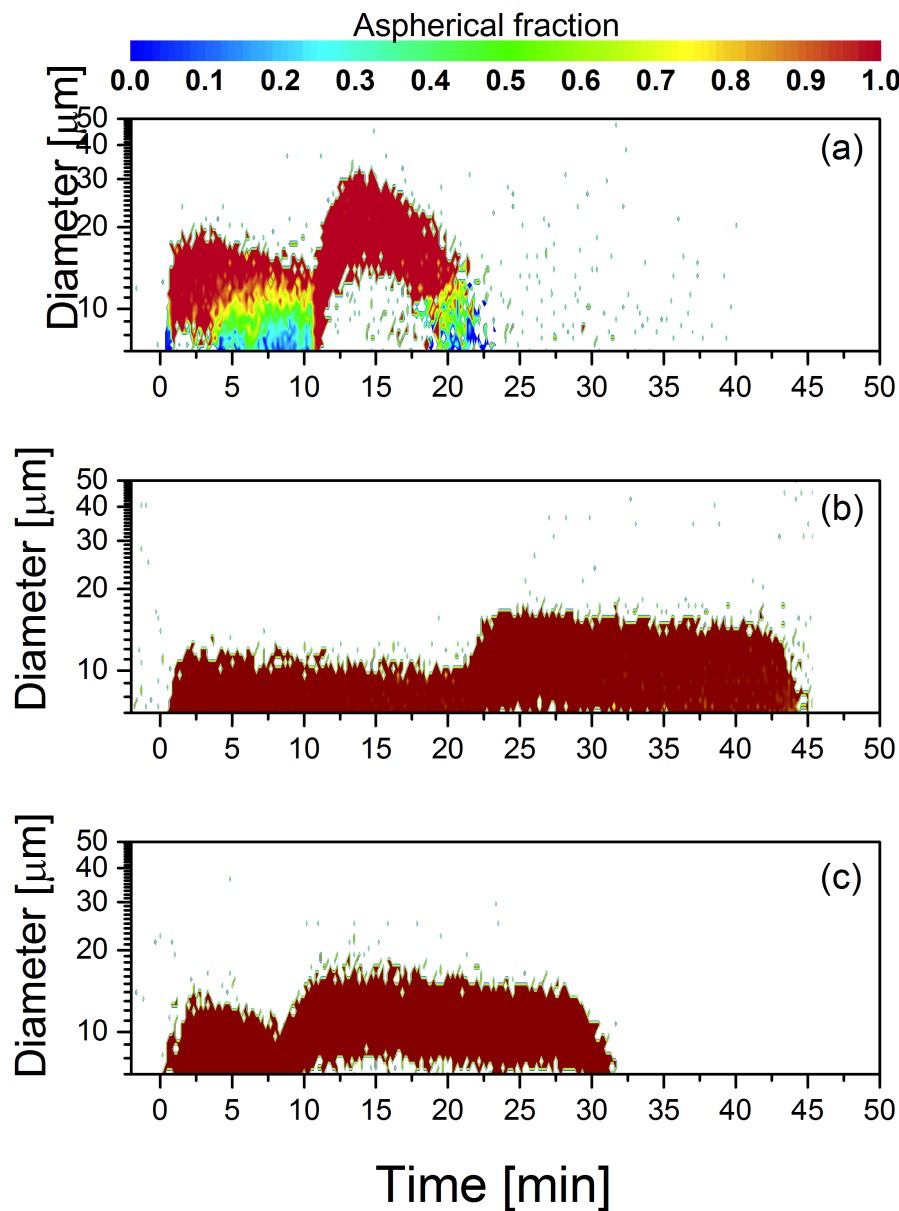

**Figure 3.** Size-segregated aspherical fraction measured by PPD-2K (see Sect. 2.3.1). (a) Run no. 1292.01 (–30 °C), (b) Run no. 1276.05 (–40 °C), (c) Run no. 1298.12 (–50 °C).





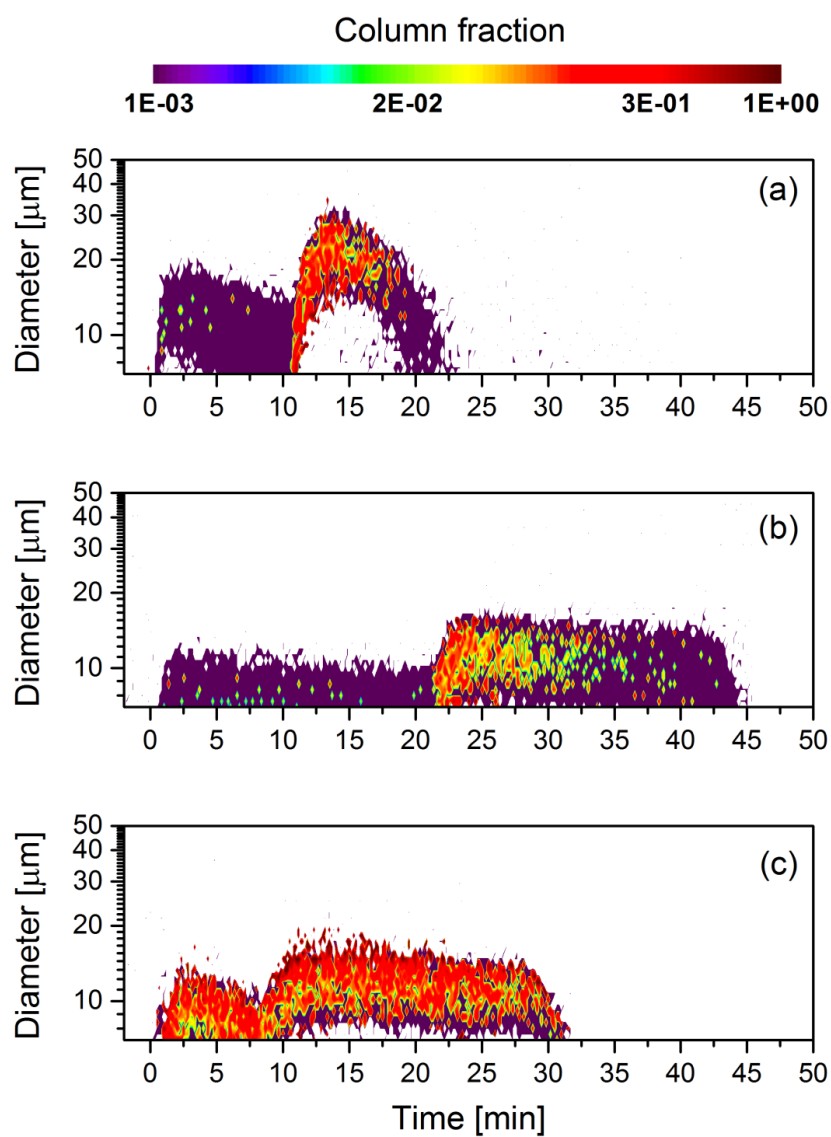

2 **Figure 4. Size segregated fraction of columns on log scale, measured by PPD-2K. (a) Run no. 1292.01 (–30 °C), (b)**

3 **Run no. 1276.05 (–40 °C), (c) Run no. 1298.12 (–50 °C).**



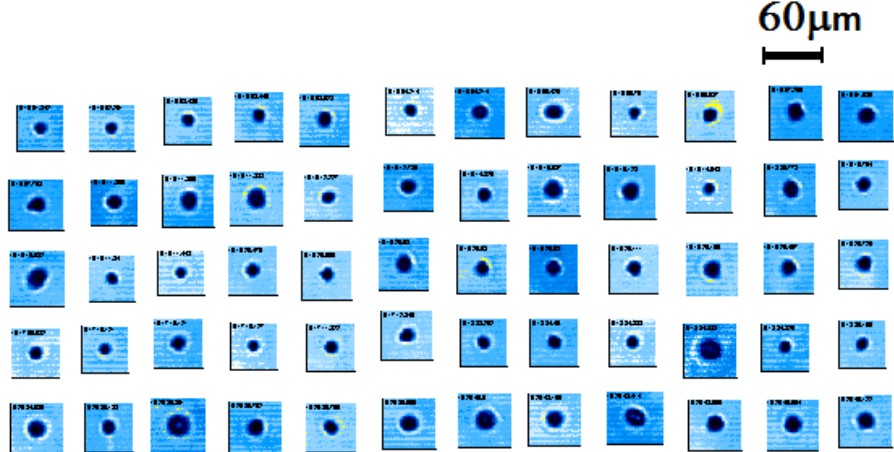

2    **Figure 5. Experiment no. 1292.01. 3V-CPI images of frozen droplets immediately after phase transition. Shape**

3    **analysis of these particles is presented in Fig.6.**

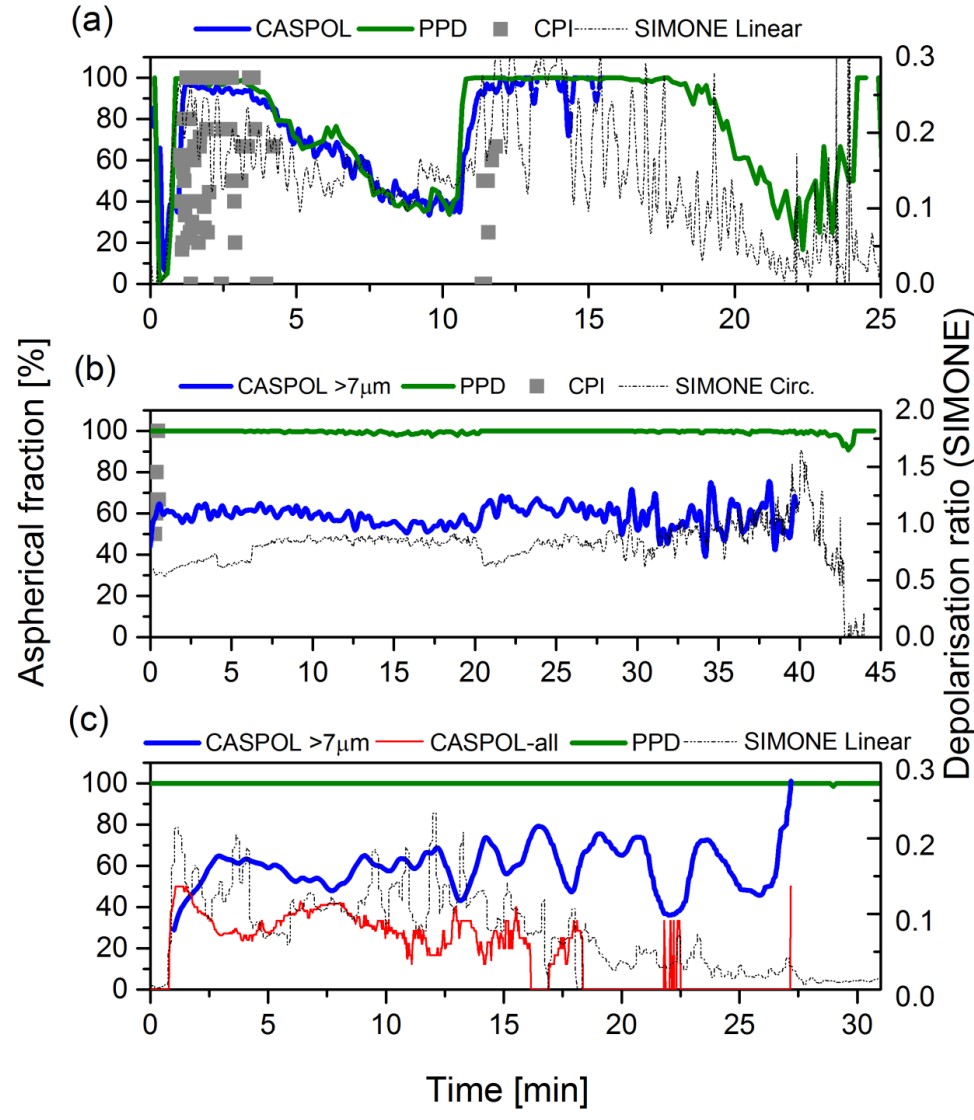

Figure 6. PPD-2K (green line), CASPOL (blue line) aspherical fraction of a subgroup of particles with diameter > 7 μm. The inter-comparison complemented by SIMONE linear or circular depolarisation ratio (dashed line), CASPOL aspherical fraction for all diameters (red line) and 3V- CPI non-round (aspherical) fraction (grey rectangles). (a) Run no. 1292.01 (–30 °C), (b) Run no. 1276.05 (–40 °C), (c) Run no. 1298.12 (–50 °C).

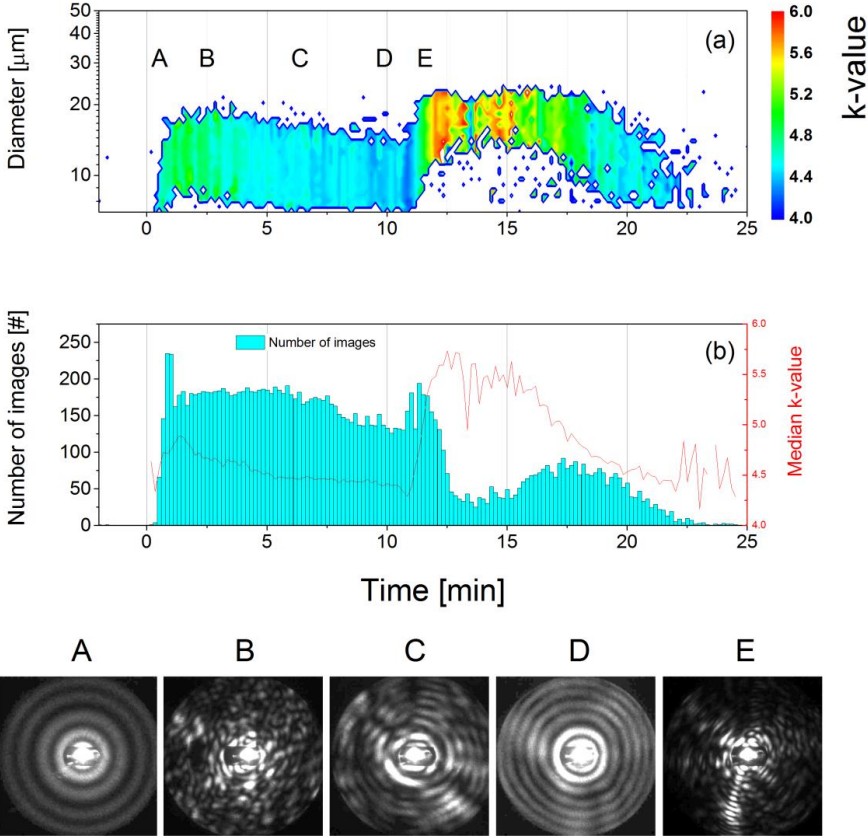

Figure 7. Evolution of small-scale complexity in experiment 1292.01 (–30 °C). (a) The size-segregated k-value (complexity parameter), capital letters correspond to the scattering patterns presented at the bottom, (b) Number of the scattering images used for the analysis and the median k-value. Lowest panel: 2-D scattering patterns from PPD-2K that have been collected during periods indicated with the letters A–E. For size segregated k-values at lower temperatures see Fig. S2.

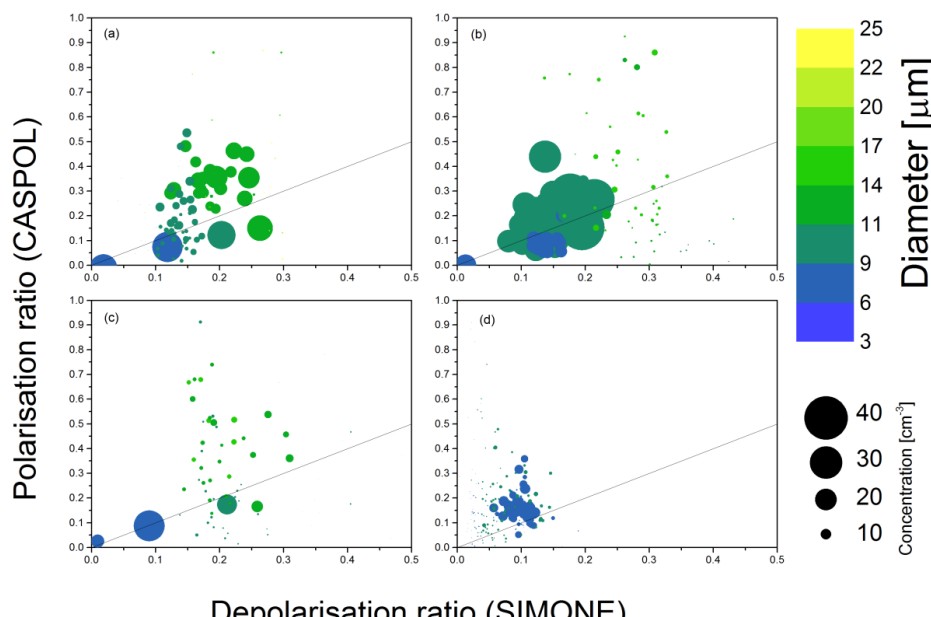

2      **Figure 8. CASPOL polarisation and SIMONE depolarisation comparison for runs (a) no. 1292.01, (b) no. 1291.07, (c)**

3      **no. 1291.12, (d) no. 1298.20, for details see Table 1. Marker size annotates number concentration, with highest at 56**

4      **cm$^{-3}$. Diameter is colour coded. Black reference line is 1:1 ratio.**