# Peer review of "Intercomparison study and optical asphericity measurements of small ice particles in the CERN CLOUD experiment"

_Atmospheric Measurement Techniques, 2016_

## Referee Comment (RC1) · Anonymous Referee #1 · 22 Aug 2016

General Comments:

The manuscript reports results of cloud chamber experiments in the Cosmics-Leaving-OUtdoor-Droplets (CLOUD) chamber at European Organisation for Nuclear Research (CERN). Three instruments are operated in conjunction with the chamber: Particle Phase Discriminator mark 2 (PPD-2K, Karlsruhe edition) were compared with Cloud and Aerosol Spectrometer with Polarisation (CASPOL) measurements and images captured by the 3View Cloud Particle Imager (3V-CPI). Averaged path light scattering properties of the simulated ice clouds were measured using the Scattering-Intensity-Measurements-for-the-Optical-detectioN-of-icE (SIMONE) and single particle scattering properties were measured by the CASPOL.

[Figure]

While the manuscript makes a worthy attempt at evaluating, comparing and contrasting the measurements from the instruments, it falls woefully short in its present form. There are many unsubstantiated claims and problems that need to be corrected (if possible) before this paper should be moved from AMTD to AMT.

1. There is constant reference to sub- and super-saturated conditions (presumably w.r.t. ice) in the chamber, yet no RHice measurements are shown, and there is no mention if instrumentation was available to make the measurement. The lack of RH measurements creates an uncertainty in assertions that the air immediately surrounding the drops is subsaturated, saturated or supersaturated w.r.t. ice (or water for that matter).

2. T and P are measured, but inferring that all of the water drops become ice is not supported by any measurements, but only inferred by the instruments that are being evaluated for their ability to discriminate ice and water drops. The measurements (Fig. 1) suggest rapid nucleation by CCN and formation and growth of particles following a rapid expansion in the chamber. It is assumed that these are water drops and that the drops immediately freeze. However, the measured temperature only drops to about -35.5 C, not as low as the homogeneous freezing temperature. Also, the depolarization ratio reaches a modest maximum of 0.25. There is no way to confirm if all of the water drops froze, or not. I would like to see a similar time series for an expansion conducted at the colder temperature (below – 40 C) where homogeneous freezing is assured.

3. There is no standard for determining whether spherical particles are water drops or ice. The PPD-2K is assumed to be capable of distinguishing spherical water from ice based on individual particle diffraction fringes, but as shown in Fig. 7, the comparison between the scattering pattern in A (water drop) and D (sublimated ice), it is not possible to unambiguously determine spherical ice from a water drop.

4. One of the conclusions stated in the Abstract is that bulk averaged path depolarisation measurements of these clouds showed higher correlation to single particle measurements at high concentration and small diameters of cloud particles. Yet, measurements of small (in this case < 7 microns) are only made by one instrument (CASPOL), and there is no way to determine why there is a (very poor) correlation (as shown in Fig. 8) and how to determine the physical significance. The statement in Section 3.4 lines 33 – 34 that the correlation in Fig. 8 is surprisingly reasonable leaves this reviewer bewildered. It looks to me like the correlation is terrible. The (max) $R^2$ value of 0.35 in regions with small particles at high concentrations (where there is no way of actually knowing the shape of the particles) is nothing to brag about, and $R^2 = 0.01$ in regions with low particle concentration is pitiful.

5. The 3V-CPI is the only instrument that provides actual images of these particles. Even though the CPI pixel size resolution is not optimum for resolving the shape of these small particles, the manuscript needs to show more images of particles. Specifically, show images of the water drops prior to freezing. Also, there is mention of columnar shapes identified by the PPD-2K, but no CPI images. Please show the CPI images that correspond with the PPD-2K derived columns. I don't understand the CPI measurements in Fig. 6. How are the gray squares calculated? Why are there multiple overlapping measurements at the same point in time? If each point represents an individual image analysis, then why weren't the other single particle measurements processed in this manner. Why are there not more CPI measurements in Figs. 6b,c?

6. There is no description of how the instruments were operated. Were the instruments installed in the cloud chamber? Was cloud air exhausted through the sample volume of the probes? Were the probes aspirated? Etc. There also needs to be more description of how the instruments were operated and how the measurements were processed. It is not straightforward how to measure the sample volume of instruments used to measure particle size distributions from a cloud chamber. How were the size distributions computed? The agreement in the size distributions shown in Fig. 2 is poor, often differing by an order of magnitude. How does this affect the results reported in the paper?

Overall, I am not sure what the takeaway messages are from this paper. None of the instruments tested are capable of unambiguously distinguishing ice from water. "Complexity" is discussed but never really defined, except to hypothesize that it is a "frost" layer. Based on the PPD-2K diffraction images, the instrument can show the difference between a spherical particle and a particle that is irregular in shape or has some surface "complexity", but there is no convincing explanation of how to apply this information quantitatively. The k value is mentioned, and in other papers there are examples of diffraction patterns from analogs and other shapes, but there is no comparison with high-resolution images of actual ice particles. The images from the PHIPS-HALO instrument in Schnaiter et al. (2016) do not have adequate resolution to provide useful information, except to distinguish columns from quasi-round particles. After looking at the diffraction patterns in Schnaiter et al. (2016) I cannot tell the difference between a distorted (analog) scattering pattern and one in this manuscript that is labeled as having surface complexity. There also appears to be no additional information on how well the diffraction patterns correlate with actual high-resolution images of ice particles in Vochezer et al. (2016). Ideally, an instrument capable of imaging particles with much higher optical resolution than the CPI should be used to compare with the PPD-2K. Could ice particles be captured on a cooled slide, placed in a cold box and photographed under a microscope? Even though the CPI only has adequate resolution to distinguish round, quasi-round and columnar shapes for particles $> \sim 30$ microns, I would still like to see a comparison between CPI images and PPD-2K diffraction patterns of the various particle shapes that are mentioned in the manuscript.

The SID family of instruments (including the PPD-2K) provide interesting and potentially useful measurements, but the quantitative utilization of these measurements in mixed-phase and in cirrus clouds with a combination of growing and sublimating particles is not clear. Measurements have shown that a substantial fraction of false irregulars are seen in all-water clouds (i.e., Johnson et al. 2014 JAS). Yes, certain pristine shapes can be identified: perfect spheres, column shapes and possibly hex shapes, but the large majority of ice particles in cirrus are irregular. How are these particles

quantified?

The ability of the CASPOL to quantitatively distinguish water and ice is not demonstrated at all. The results vary with both particle size and concentration, leaving one to wonder what it is really measuring. There is good qualitative agreement with the PPD-2K in estimating asphericity in Fig. 6a, but no agreement in Figs. 6b and 6c. What is the explanation for this?

Specific Comments:

P. 2 Lines 19 – 20: I disagree. Shape is used more often than scattering intensity in mixed-phase clouds, and arguably more reliably. In many cases in mixed-phase (i.e., water saturated) clouds, ice particles rapidly grow to sizes where they can be distinguished from water drops using CPI imagery (see Lawson et al. 2015 – JAS).

P. 2 Lines 32 – 32: The measurement of particles smaller than 50 micron using the FSSP were contaminated with shattering. Delete this reference.

P. 4 Lines 1 – 3: "We then use the asphericity to determine the ice fraction in a cloud by prescribing an aspherical shape for all the ice particles, and hence assume that ice fraction is equivalent to an aspherical fraction." As discussed above, using the measurements presented in this manuscript, there is no way to unambiguously determine if asphericity explicitly distinguishes ice particles from water drops. This statement needs to be modified or deleted and then explained later in the text after it is understood that using asphericity is an estimate of ice fraction that is not well quantified under all conditions.

P. 4 Lines 10 – 14: This statement appears to be contradictory. If LWC is independent of updraft velocity, but stronger updrafts produce a higher concentration of smaller drops, which then freeze, how is IWC increased in stronger updrafts? This appears to violate conservation of mass.

P. 5 Line 6: 100 per cc is not necessarily a low concentration. Simulations now show

that coincidence occurs at this concentration with the CASPOL and multiple scattering will occur in ensemble measurements. Please qualify this statement (and not by using 1980's references to the FSSP).

P. 5 Lines 26 – 30: Please show some quantitative evidence that 10-5 asphericity threshold actually applies to ice/water discrimination. Otherwise, please state that this is a subjective value based on visual analysis of the scattering pattern. Referencing Vochezer et al. (2016) is not sufficient.

P. 6 Lines 1: Detection of a bulk cloud phase is meaningless unless the cloud is all-water (T > 0 C), or known to be all-ice (i.e, colder than – 40 C). There is no quantitative information published (yet) on bulk measurements of the ice fraction in mixed-phase.

P. 6 3V-CPI: The references in this section are terrible, misleading and in one case unavailable. The 2D-S portion of the 3V-CPI should be referenced by Lawson et al. (2006) – JTech. The CPI portion of the 3V-CPI should be referenced by Lawson et al. (2001). Lawson et al. (2003) should be deleted. The Heymsfield et al. (2010) reference does not show particle habit classification schemes. This reference should be replaced by, for example, Lawson et al. (2006) – JAMC; Um and McFarquhar (2009) – QJRMS; Lindqvist et al. (2012) – JGR

Section 3.1.1: As explained above, there are way too many assumptions about what is happening during the first rapid expansion and for a few seconds or minutes afterward. How do we know that all of the drops froze instantaneously? Could there be a mixed-phase cloud and Bergeron process occurring after the rapid expansion?

P. 8 Line 16: How do you know there was no coincidence?

P. 12 Lines 1 – 9: There are several assumptions and generalizations in these lines that need to be deleted based on previous arguments in this review.

Conclusions: This also needs to be re-written to tone down all of the claims that are not substantiated. BTW – the measurements presented in this manuscript extend out

to particle sizes of about 30 microns. On line 26 and another place in the manuscript the claim is that the results are valid out to 60 microns. Where does this come from?

---

## Referee Comment (RC2) · Anonymous Referee #3 · 31 Aug 2016

**1   Comments**

This study attempts to understand observations of small ice and liquid in simulated clouds at the CERN-CLOUD (European Organisation for Nuclear Research - Cosmics-Leaving-OUTdoor-Droplets ) chamber, using laboratory instruments PPD2K (Particle Phase Discriminator mark 2 - Karlsruhe) and SIMONE (Scattering Intensity Measurement for the Optical detectioN of icE) and airborne instruments CASPOL (Cloud and Aerosol Spectrometer with Polarisation) and 3V-CPI (3 View Cloud Particle Imager).

There is a great need to understand this type of measurement and so the aims of this paper are highly relevant at this time.  What the work does do is to highlight the

difficulty, even in the controlled conditions in the cloud chamber, of making the distinction between spherical ice and liquid particles, and this is valuable, especially when presented for a relatively new instrument like PPD2K.

The implications of these difficulties are not properly explored though, and if this were expanded the study would benefit. There is no real path forwards presented and little guidance as to where and when each type of measurement can offer clear advantages. The PPD2K seems to offer benefits, but the chance to fully exploit this measurement is not taken here, for example - the calibration of surface complexity has not yet been performed. Work to fully characterise the small scale complexity would help, but the instruments used for comparison (CASPOL, 3V-CPI), as stated in the work, do not have sufficient resolution, which makes the aim difficult.

The work is somewhat over ambitious in its claims and is rather unfocussed. Is this a comparison between different ice formation situations (liquid vs. in situ), or is it a single particle instrument comparison, or is it a single particle vs. bulk averaged measurements comparison? It is in part all of these, but with the result that not one area is explored in that much depth. If this work is to be a comparison of different ice formation scenarios then the details of the in situ cases are hidden away in the supplementary material somewhat. If the work is to be a detailed instrument comparison then there needs to be more discussion about the instruments and processing, e.g. phase discrimination and techniques and thresholds. Both the abstract and conclusions sections should be thinned out to contain only the firm statements of work done and the robust conclusions or the claims supported by stronger evidence. If this is done and the following recommendations are accounted for then the work should be promoted to AMT.

**2  Specific sections**

There is no mention of small hexagonal ice after the abstract.

**2.1 Section 1 - Introduction**

There is no mention here, or elsewhere, as to on what scale the small-scale complexity is expected to be present for atmospherically relevant particles. This is required for discussion of the PPD2K and comparison with the resolution of probes as it is mentioned often in the text. Mentioning the sizes and concentrations ranges of particles in the clouds (line 24) would be helpful here, especially when referring to remote sensing limitations (size / wavelength dependant).

**2.2 Section 2 - Methodology**

The paper tends to present the work in terms of the -30 degree C case, the liquid to ice transition case, and this is good because understanding this the phase transition is crucial, and poorly observed in the real atmosphere. Complimentary measurements at -40 degree C and -50 degree C are presented. It would be good to mention the difference between liquid origin and in situ cirrus, e.g. Kramer et al. A microphysics guide to cirrus clouds - Part 1: Cirrus types, ACP, 2016.

The two cases are not well described, neither is the motivation for doing the two types of expansion. The information for the deep convection (-30 degree C?) and in situ (-40 and -50 degree C) cases could be added to Table 1, and more discussion given to the differences. Crucially, how does supersaturation evolve over time in these simulations, especially with regard to liquid supersaturation in the -30 degree C case, and the ice-subsaturation in all cases, for sublimation. A presentation of the cooling rate as well as the measured temperature would help here and additionally, what are the equivalent updraughts and are these reasonable.

The abstract claims to measure the response of four probes, but the reasons for the differences between the probes, e.g. technique, sample volumes, wavelengths, collection angles, are not explored in much detail. Also there is no information on how

the airborne and PPD2K probes were aspirated, and what flow rates and particle rates were encountered. It would be good to provide information on the numbers of particles per second that the probes encountered so that coincidence can be ruled out, and a comparison against what they are designed for, when fitted to an aircraft. There is a brief discussion of coincidence, but no evidence that it was not present.

- page 4 line 9 - this subject is a good exploitation of the additional cooling rate available in this particular chamber

- page 4 lines 11 to 14 - There seems to be a mismatch between consistent frozen mass (IWC) and smaller, more numerous particles, or higher IWC.

- page 4 line 12 - Ackerman 2015 has now been moved from ACPD to ACP

- page 5 line 7 - the CCN and cloud particle number data in Table 1 seem to disagree with the words, that all CCN are activated at low concentrations. What about high concentrations? And how is Table 1 ordered?

- page 5 line 11 - Referring to the figure and the supplementary material would be helpful here. Including the supersaturation in the figures should also be done if this information is available.

2.3   Section 3 - Results and Discussion

This section is a fairly complicated read, and is a mixture of instrument artefacts, experimental design, and results. Some parts may be better in the instrument section (methodology), and others in a separate experimental design followed by results section. For example discussion of coincidence errors (e.g. page 8 lines 13-16) and the 3V-CPI discussion in section 3.2.

It isn't clear how it is known that the supercooled liquid regime only lasts a few seconds, and if the phase change and temperature changes happen uniformly throughout the chamber or not. Is time zero defined when water saturation is reached? Do the observations confirm the pathway for quasi-spherical ice formation as claimed? The presentation of results here doesn't make this clear. What specifically does "complex particles" refer to on page 8 line 10.

Section 3.3 - The needs to be specific mention of which probes are being compared where. The CPI data as presented here look very variable, and it is difficult to infer trends in particle asphericity.

The analysis of the SIMONE and CASPOL data in figure 6c is possibly accurate, but very difficult to assess on the time series (page 9 line 21). The early period and late period look as though they might have different behaviour that a more detailed analysis would confirm or refute.

It seems in figure 7a that the particles all freeze at a similar time, independent of size, which seems like an important observation that warrants a more detailed discussion regarding the implications for atmospheric clouds. There does not seem to be evidence presented here that the particles develop a frost layer - is this assumption just based on Jarvinen 2016c? Do they remain liquid in the centre for longer? And is droplet shattering during freezing important - is number constant? I can't see any size segregation in figure 7b that suggests the smaller particles are less complex - can some example scattering patterns illustrate this?

The limitations on CASPOL in phase determination are important and so this is a valuable observation - is it possible to put numbers on this, both size and concentration?

Section 3.4 - The section comparing SIMONE and CASPOL is good for completeness of the study, but limited in impact. What are the effects of the difference in sample volume / path length, wavelengths, scattering angles and how does this impact the comparison? The should be more discussion on the implications of this comparison for

real atmospheric measurements. There is no real quantitative analysis, and no limits or thresholds, in terms of size or concentration, that specify when a comparison might hold or fail. The correlations look weak in all cases. Would it help to average over a longer time period than 1 s?

Section 3.5 - This section should be expanded and more made of what the observations presented in the chamber mean for real experiments making measurements in the atmosphere, for example - what will happen at aircraft speeds to these techniques? It seems as though the main take-home messages of the work should be in this section.

- page 12 line 3 - how high concentrations - which way does the error go, higher or lower fraction?

- page 12 line 6,7 - which techniques, single particle and ensemble measurements?

- page 12 line 8,9 - Is it possible to specify ranges of size and concentration, or thresholds where the two techniques are comparable

**2.4 Section 4 Conclusions**

The conclusions as presented are useful, in that there is evidence presented that show the challenges to the atmospheric measurement community. However it would be good to see a quantitative assessment of when the probes can and can't be used for what purpose. Increasing resolution in future probes may help, but how much further given the optical limits. Also, despite the limitations the current probes need to be exploited, but there is no clear message on how to do this. The conclusions refer to particles less than 60 microns, although CASPOL only measures up to 50 microns and particles in the study are smaller than this. It is not clear how these results will apply to other probes, especially impactors. Future probes may have better resolution detection, but

depending on the size of surface complexity the limit could be the optical wavelengths used.

**3  Typographical errors**

- page 3 line 19 - 'a' climatic impact, or climatic importance?
- page 3 line 24 - cases plural?
- page 9 line 10 - ice fraction is referred to as aspherical fraction elsewhere
- page 9 line 13 - is this from 19 min onwards?
- page 9 line 31 - replace to with of
- page 9 line 32 - with respect to
- page 10 line 36 - start a new paragraph?
- page 10 line 28 - its, no apostrophe
- page 10 line 33 - patterns
- page 11 line 22 - averaging deviation
- page 11 line 21 - capitalisation of lowest
- page 11 line 37,38,39 - suggest rewrite for clarity
- page 11 line 27 - Implications "For"....
- page 12 line 36 - is the comma required?

[Figure]

**4   Figure comments**

- figure 3 - font size of axis titles

- figure 6 - linear or circular?

- figure 7b - the complexity line is very faint and hard to see

- figure 8 - difficult to read and extract the take home message, especially when there are lots of high concentrations data points. Font sizes are all different.

---

## Author Comment (AC1) · 17 Nov 2016

We would like to thank the Referees for for their comments that improved the quality of the paper. Please find the Author Comments and replies to the individual issues specified by the Referees, the annotated paper and the supplement in the uploaded zip file.

Please also note the supplement to this comment: http://www.atmos-meas-tech-discuss.net/amt-2016-205/amt-2016-205-AC1-supplement.zip